# Video Token Merging
# for Long-form Video Understanding

**Seon-Ho Lee**[*]
Korea University
seonholee@mcl.korea.ac.kr

**Jue Wang**
Amazon AGI
juewangn@amazon.com

**Zhikang Zhang**
Amazon AGI
zhikang@amazon.com

**David Fan**[†]
Meta FAIR
davidfan@meta.com

**Xinyu Li**
Amazon AGI
xxnl@amazon.com

## Abstract

As the scale of data and models for video understanding rapidly expand, handling long-form video input in transformer-based models presents a practical challenge. Rather than resorting to input sampling or token dropping, which may result in information loss, token merging shows promising results when used in collaboration with transformers. However, the application of token merging for long-form video processing is not trivial. We begin with the premise that token merging should not rely solely on the similarity of video tokens; the saliency of tokens should also be considered. To address this, we explore various video token merging strategies for long-form video classification, starting with a simple extension of image token merging, moving to region-concentrated merging, and finally proposing a learnable video token merging (VTM) algorithm that dynamically merges tokens based on their saliency. Extensive experimental results show that we achieve better or comparable performances on the LVU, COIN, and Breakfast datasets. Moreover, our approach significantly reduces memory costs by **84%** and boosts throughput by approximately **6.89** times compared to baseline algorithms.

## 1   Introduction

Over the past few years, the Transformer architecture (Vaswani et al., 2017) has risen as a revolutionary paradigm within natural language processing (NLP) (Devlin et al., 2018) and has seamlessly expanded its influence into the domain of computer vision (Wang & Torresani, 2022; Bertasius et al., 2021; Wang et al., 2022; Akbari et al., 2021; Li et al., 2022; Fan et al., 2021). This expansion has been exemplified by remarkable achievements in recent multi-modality foundation models such as Sora (Brooks et al., 2024), GPT4 (Achiam et al., 2023), and Gemini (gem), showcasing its exceptional performance and versatility across diverse applications.

In contrast to the natural language processing, the visual input has much lower information density and thus tokenizing the raw RGB image as non-overlapped patches becomes the essential operation in vision transformers (Wang & Torresani, 2022; Bertasius et al., 2021; Dosovitskiy et al., 2021). However, the computational cost of transformer exponentially increases with the sequence length, which generates tremendous computation when feeding visual input into the large-scaled transformer models with billions of parameters. Due to the redundancy in video sequence, this phenomenon becomes more severe with video input, especially for long-form videos. This bottleneck impedes the

---

[*]Work done during an internship at Amazon Prime Video.
[†]Work done while at Amazon Prime Video.

38th Conference on Neural Information Processing Systems (NeurIPS 2024).

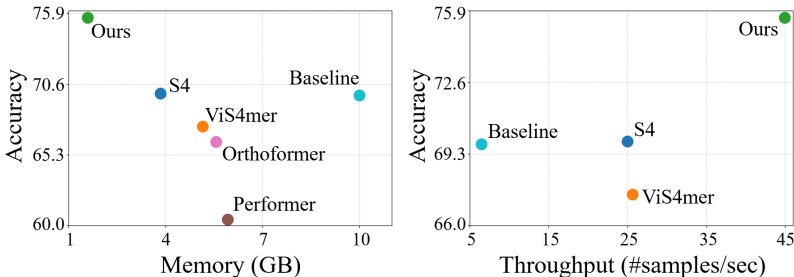

Figure 1: Comparison of GPU memory footprint and throughput against scene prediction accuracy on the LVU dataset (Wu & Krähenbühl, 2021).

further advancement of foundational models in handling long-form video data. Various attempts (Yin et al., 2021; Wang et al., 2021; Meng et al., 2022; Rao et al., 2021; Liang et al., 2022) have been proposed to improve the efficiency of vision transformer by introducing a token selection module. However, these methods are primarily designed for images and may require non-trivial adaptation to the long-form video scenarios due to the video-level long-term dependencies and motion dynamics. Moreover, tokens dropped by the token selection module cannot be reused in later layers, which may result in the loss of important information.

In addition to the token selection, token merging techniques (Bolya et al., 2022; Ren et al., 2023; Bolya & Hoffman, 2023; Li et al., 2024) have been proposed to increase the efficiency and effectiveness of transformer-based networks. Specifically, they reduce the sequence length by merging similar tokens, thereby decreasing the computational cost. In addition to the efficiency, token merging demonstrates huge advantages by increasing the contextual information so that the model can learn from patterns presented across multiple tokens. Previous token merging algorithms in both the image and video domains use manually designed token partitioning methods and merge tokens based on their similarity. Even though the merged tokens would still keep the original information, they may have different granularity after the merging operation. In this paper, it is argued that different regions in the visual data may have different information density. Since the discriminative information of tokens may be degenerated after merging, some visual tokens should not be merged even if they look similar to each other. Rather than relying solely on similarity, we question whether more unmerged tokens should be used to describe salient areas, while merging more tokens for the background.

In this paper, we explore various video token merging (VTM) methods in long-form video classification task and aim to find the effective token merging method for long-form videos. Previous video token merging method (Li et al., 2024) only decouples the spatial and temporal dimensions, which is unfavorable, especially for long-form videos. As the long-term dependency plays an important role in the long-form video understanding, spatiotemporal visual tokens should be considered jointly. Sequentially merging token along with one dimension after another may generate biased prior. In our work, we first naively extend the image-based token merging (Bolya et al., 2022) to the video domain and then propose region-centralized and motion-based token merging algorithms, which estimate the salient region of video sequences. Finally, we develop a learnable VTM which predicts the saliency score of each token and adaptively merges spatiotemporal visual tokens in data-driven manner. Experimental results demonstrate that the proposed algorithm improves the effectiveness and the efficiency of the transformer-based network and outperforms the conventional long-video understanding methods with better throughput and less memory usage, as also shown in Figure 1.

We summarize the contributions of this paper as following:

- We explore various video token merging methods including the naïve VTM, the region-concentrated VTM, and the motion-based VTM.

- We propose the learnable video token merging algorithm, which estimates the saliency score of each token and adaptively merges visual tokens based on their scores.

- The proposed algorithm achieves the best or competitive results on various datasets including LVU, Breakfast and COIN. Moreover, we significantly reduce memory costs by 84% and improve the throughput by 6.89 times via the proposed learnable VTM.

## 2 Related Work

### 2.1 Long-form Video Modeling

Transformers have demonstrated remarkable prowess in capturing long-term dependencies, as evidenced in their success in natural language processing (NLP) tasks (Brown et al., 2020; Dai et al., 2019; Devlin et al., 2018). However, the intensive computational requirements stemming from dense self-attention calculations (Vaswani et al., 2017) pose a significant obstacle not only in NLP but also in the domain of computer vision, especially for the long-form videos. Many recent video transformer works (Wang & Torresani, 2022; Liu et al., 2021; Bertasius et al., 2021) focuses on improving the global attention mechanism. However, they are not designed for dealing with redundant spatial and temporal image tokens that are common in long-form video scenarios. To capture longer temporal information, LF-VILA (Sun et al., 2022) develops a hierarchical architecture to include more frames in the model. Similarly, MeMViT (Wu et al., 2022) utilizes longer temporal information by emerging the previously cached "memory" from the past. A novel alternative to transformers is the Structured State-Space Sequence (S4) model proposed by Gu et al. (2021), which models the long-range dependencies by simulating a linear time invariant (LTI) system. Subsequently, ViS4mer (Islam & Bertasius, 2022) and S5 (Wang et al., 2023) extend S4 model to the long-form video classification task. ViS4mer (Islam & Bertasius, 2022) stacks multiple S4 layers with different scales in modeling long-form videos, and S5 (Wang et al., 2023) include an additional selective module to further improve the performance. Unlike these works that focus on the improvement of architecture and attention mechanism, this paper will start from a more basic concept in the transformer, video tokens and how to effectively merge them. Even though our proposed method can theoretically be applied on S4 model, the scope of this paper is on the well established transformer architecture. We will leave the investigation of video token merging on S4 (Gu et al., 2021) in the future work.

### 2.2 Adaptive Token Selection

Adaptive token selection is widely used to improve model efficiency by leveraging a light-weight selection module to pick up the 'useful' tokens while dropping the 'unuseful' ones. In vision transformer, STTS (Wang et al., 2021) utilizes a token selection module known as the named scorer network to assign importance scores to each token, subsequently selecting the top-K frames with the highest scores. Building upon this concept, AdaViT (Meng et al., 2022) further extends the approach by developing instance-specific policies. These policies guide the activation of patches, self-attention heads, and transformer blocks, enhancing adaptability and efficiency in processing visual data. STTS, AdaVit and other similar approaches (Wang et al., 2021; Meng et al., 2022; Rao et al., 2021; Liang et al., 2022) drop a significant number of tokens in the early decision stage to save more cost, but the dropped tokens cannot be reused in the later layers, which is easier to degenerate the contextual information in the long-form videos.

### 2.3 Token Merging

Visual token merging is first proposed in (Bolya et al., 2022) which aims at increasing the throughput of existing ViT models without training. Following works (Ren et al., 2023; Bolya & Hoffman, 2023; Li et al., 2024) leverage this idea to save computational cost in different downstream applications, such as diffusion model, video and language understanding, and video editing. Specifically, visual tokens are first partitioned into two sets with equal size; for each of the edge tokens in one set, find the most similar token in another set and merge them by average pooling; finally, concatenate two sets back together. Although the token merging is simple and effective, its applications have mostly remained in the image domain. There is no fundamental research work has been explored for the long-form video token merging strategies, where the spatiotemporal tokens are more redundant and embed complicated dependencies locally and globally. In this work, we argue that visual tokens from the long-form video should be carefully partitioned and merged based on the salient areas in videos. To this end, we ablate various video token merging algorithms and provide extensive expermental results and analysis.

## 3 Proposed Algorithm

### 3.1 Preliminary – Token Merging

Token merging (Bolya et al., 2022) aims to reduce the redundancy by merging similar tokens at each transformer block, thereby increasing the effectiveness and efficiency of a transformer-based network. Specifically, token merging has three steps: partitioning, matching, and merging.

**Partitioning:** For given a set of $N$ tokens $\mathcal{X} = \{x_1, x_2, \ldots, x_N\}$, token merging first partition $\mathcal{X}$ into a set of target tokens $\mathcal{T}$ and a set of source tokens $\mathcal{S}$, given by

$$\mathcal{T} = \{x_i : i \bmod \gamma = 0\}, \tag{1}$$
$$\mathcal{S} = \{x_j : j \bmod \gamma \neq 0\} \tag{2}$$

where $\gamma$ is partition factor and $\bmod$ denotes the modulo operator. Thus, $|\mathcal{T}| = \frac{|\mathcal{X}|}{\gamma}$. Also, $\mathcal{X} = \mathcal{T} \cup \mathcal{S}$ and $\mathcal{T} \cap \mathcal{S} = \varnothing$.

**Matching:** Then, for each source token in $\mathcal{S}$, it finds the most similar target token in $\mathcal{T}$. Here, the similarity between tokens $x_i$ and $x_j$ is defined as the cosine similarity of the corresponding key vectors $k_i$ and $k_j$, which are obtained in the most recent self-attention layer. For a source token $x_j \in \mathcal{S}$, the index of its matched target token $m_j$ is obtained by,

$$m_j = \operatorname*{argmax}_{i:\{i:x_i \in \mathcal{T}\}} \frac{k_i^t k_j}{|k_i||k_j|}. \tag{3}$$

**Merging:** Lastly, token merging merges the tokens based on the matching results. For each target token $x_i \in \mathcal{T}$, it obtains the merged token $y_i$ by using average pooling,

$$y_i = \frac{x_i + \sum_{j \in \mathcal{M}_i} x_j}{1 + |\mathcal{M}_i|} \tag{4}$$

where $\mathcal{M}_i = \{j : m_j = i, \forall x_j \in \mathcal{S}\}$ is the index set of source tokens which are matched with $x_i$. In this case, the number of tokens is reduced by $|\mathcal{S}|$ after token merging. It can also control the number of reduced tokens by $R$ by reassigning the matching index as

$$m_j = -1 \tag{5}$$

for all $x_j$ except for the source tokens with the $R$ highest similarity scores.

### 3.2 Problem Definition

Suppose that a video with $L$ frames is given, where $L \geq 60$. To perform the classification or regression on the given video, we can use a simple transformer-based network, which is shown in Figure 2 (a) and (b). It first obtains the tokens $X_1, X_2, \ldots, X_L \in \mathbb{R}^{H \times W \times C}$ by using an encoder. Here, $H, W,$ and $C$ denote the height, the width, and the channel dimension of the token tensor, respectively. Then, it utilizes transformer blocks to update the tokens. As its input, $i$-th transformer block takes the tokens corresponding to $L_i$ frames without overlapping, where $L_i \leq L_j \leq L$ for $i < j$. The prediction head yields the estimation results. However, it requires the prohibitively large memory and computation costs due to the quadratic complexity of the self-attention $\mathcal{O}(L^2 H^2 W^2 D^2)$, where $D$ is the dimension of key vectors. Hence, our goal is to increase the efficiency of this baseline network by reducing the number of tokens via token merging, while maintaining or even improving the performances of the network by removing the redundant or noisy information in the video. To this end, we explore various token merging methods for long video processing.

### 3.3 Video Token Merging – Exploration

**Naïve video token merging:** First, we combine the standard token merging with the baseline network as intact as possible. To this end, we substitute transformer blocks with VTM blocks, in which token merging layer is inserted after the dropout layer, as illustrated in Figure 2 (c). In this naïve VTM, we employ the standard token merging in (1)-(4). At each $i$-th VTM block, naïve VTM reduces the $R_i$ tokens. For example, at $\gamma = 4$ and $R = |\mathcal{S}|$, it gradually removes the $68\%$ of tokens over the network. Hence, the computation cost of the self-attention is reduced from $\mathcal{O}(N^2 D^2)$ to $\mathcal{O}((\frac{N}{\gamma^{i-1}})^2 D^2)$ at $i$-th VTM block. As shown in Table 1, this naïve VTM shows better scores than the baseline network, because the token merging reduces the information redundancy in the videos.

**Region-concentrated video token merging:** Compared to an image, a video contains redundant spatiotemporal tokens. Depending on the tasks, some tokens are more important than others. However, as shown in Figure 3 (a), naïve VTM selects every $\gamma$-th token as the target tokens since it exploits uniform token partitioning in (1). Also, for token merging, it purely relies on the similarity between tokens regardless of the semantics, and thus the self-attention can more easily swayed by unnecessary

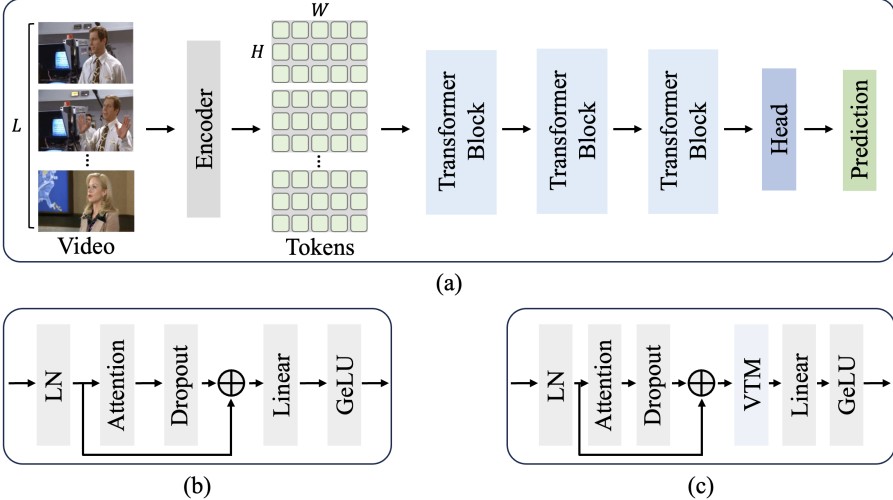

(a)

(b)                                        (c)

Figure 2: The architectures of (a) the baseline network, (b) the transformer block, and (c) the video token merging block.

Table 1: Comparison of different VTM methods on the LVU dataset. The best results are boldfaced and the second-best ones are underlined.

| Algorithm | Content (↑) | | | Meta data (↑) | | | | User (↓) | |
| --- | --- | --- | --- | --- | --- | --- | --- | --- | --- |
| | Relationship | Speaking | Scene | Director | Genre | Writer | Year | Like | View |
| Baseline | 57.14 | 36.68 | 69.76 | 62.61 | 56.73 | 49.40 | 39.86 | 0.28 | 4.18 |
| Naïve | 61.90 | 36.18 | 72.09 | 67.28 | 55.12 | 51.19 | 44.75 | 0.28 | **4.01** |
| Boundary | 59.52 | 37.18 | 69.76 | 61.68 | 57.21 | 50.0 | 47.55 | 0.26 | 4.16 |
| Center | 61.90 | 40.20 | 74.41 | 62.61 | 58.81 | 51.19 | 44.05 | 0.25 | 4.11 |
| Motion | **64.28** | 37.68 | 74.41 | 64.48 | 58.49 | **55.95** | 47.55 | 0.24 | 4.13 |
| Learnable | **64.28** | **42.12** | **75.58** | **70.09** | **59.77** | 53.57 | **48.55** | **0.21** | **4.01** |

information. Therefore, for better video token merging, it is important to consider the saliency of each token before merging them.

To investigate this issue, we explore center-concentrated video token merging and boundary-concentrated video token merging. The center-concentrated token merging samples $50\%$ of the entire target tokens from the center area with the size of $\frac{H}{2} \times \frac{W}{2}$, which uses more unmerged tokens to describe center area and merge more token from the boundary. Specifically, we use the partition factor of $\frac{\gamma}{2}$ for the center area and $\frac{3}{2}\gamma$ for the remaining area. On the other side, we implement the opposite operation for the boundary-concentrated video token merging. As shown in Table 1, center-concentrated VTM shows better performances than naïve and boundary-concentrated VTM in general. Since meaningful objects and motions are typically center-concentrated, this suggests more tokens from salient regions should be unmerged while more of the rest tokens should be merged. Figure 3 (b) shows the partitioning results of center-concentrated VTM.

**Motion-based video token merging:** Even though center-concentrated VTM has shown better performances than naïve VTM, the meaningful tokens are not always located in the center area. Moreover, the hand-crafted partitioning method forces the center-concentrated VTM to select the same number of target tokens from each frame, which is not flexible enough when applied at scale. Therefore, we explore motion-based video token merging which divides tokens into $\mathcal{T}$ and $\mathcal{S}$ based on the motion information since the moving objects carry important cues in general (Fan et al., 2023).

Let us assume that we have the magnitude of motion vector $v_i$ for each token $x_i$. We compute the sampling probability by using softmax

$$p_i = \frac{e^{v_i}}{\sum_{j=1}^{N} e^{v_j}}. \tag{6}$$

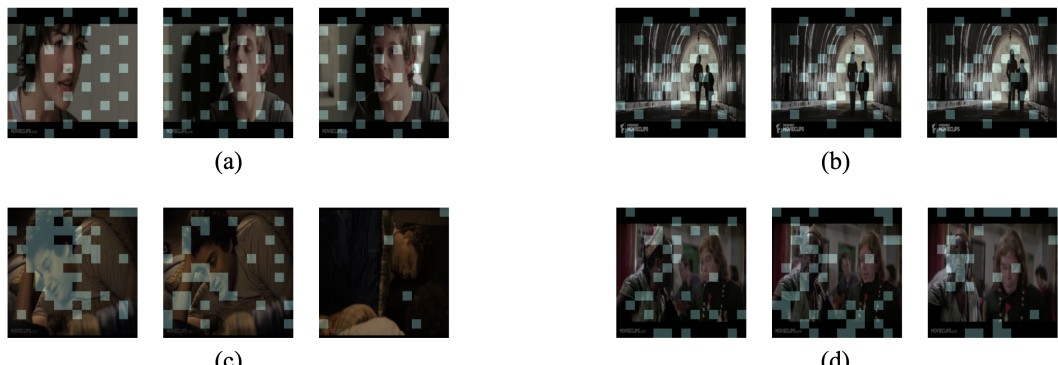

Figure 3: Visualizations of target tokens of different VTM methods: (a) naïve VTM, (b) center-concentrated VTM, (c) motion-based VTM, and (d) learnable VTM. In (d), learnable VTM selects the target tokens around salient objects rather than backgrounds.

Note that the sampling probability is proportional to the motion magnitude. Then, we construct $\mathcal{T}$ by sampling $\frac{N}{\gamma}$ tokens from $\mathcal{X}$ with the sampling probability $p_i$ for each token $x_i$.

The goal of VTM is to increase the efficiency of transformer-based network for long video understanding. Therefore, the motion information should be obtained with negligible time and computation costs. Hence, instead of estimating the motion information with an additional module, we use the motion information which is already stored in the video files; most modern video codecs, such as MPEG-4 (Richardson, 2004), H.264 (Richardson, 2004), and HEVC (Wien, 2015), exploit the motion information for efficient compression. The motion decoding takes only 0.3 milliseconds for each frame which is negligible. Figure 3 (c) shows the token partitioning examples of motion-based VTM.

### 3.4 Learnable Video Token Merging

There are some videos in which unimportant objects have large motions due to various factors such as camera movement. Motion-based VTM may not yield good results on those videos. To maximize the generalizability, we develop learnable video token merging method. Instead of depending on the motion information, learnable VTM estimates the saliency score of each token and samples the target tokens based on the estimated scores. Figure 4 shows the architecture of learnable VTM block.

Learnable VTM block contains two forward paths: a main path and an auxiliary path. Let us assume that we have a tensor of $N$ tokens $X \in \mathbb{R}^{N \times C}$. In the main path, we first obtain query $Q$, key $K$, and value $V$ by

$$Q = XU_q, \qquad K = XU_k, \qquad V = XU_v \tag{7}$$

using learnable projection matrices $U_q, U_k, U_v \in \mathbb{R}^{C \times D}$. We perform the standard self-attention on $Q, K$, and $V$ and yield the updated tokens $X'$ as

$$X' = \frac{\text{softmax}(QK^\top)}{\sqrt{D}} V. \tag{8}$$

Also, from $K$, we estimate the saliency scores $S$ of tokens by

$$S = [s_1, s_2, \ldots, s_N]^\top = \tanh(KU_s) \tag{9}$$

where $U_s \in \mathbb{R}^{D \times 1}$ is a learnable matrix. Also, $s_i \in (-1, 1)$ for $1 \leq i \leq N$. Then, we compute the sampling probability using (6) with $s_i$ instead of $v_i$ for each token $x_i$ and sample $T$ with the sampling probability as in motion-based VTM. After the token partitioning, we match and merge the tokens in $X'$ by using (3) and (4), respectively.

However, the learnable matrix $U_s$ in (9) can not be trained only with the main path since the partitioning process is not differentiable. To handle this issue, we employ the auxiliary path. This path consists of a saliency guided self-attention layer and a merging operation. The auxiliary path takes a tensor of auxiliary tokens $X_{\text{aux}} \in \mathbb{R}^{N \times C}$ as its input. Similar to the main path, we obtain query $Q_{\text{aux}}$, key $K_{\text{aux}}$, and value $V_{\text{aux}}$. Then, we perform the saliency guided attention to obtain the updated auxiliary tokens $X'_{\text{aux}}$ as

$$X'_{\text{aux}} = \frac{\text{softmax}(Q_{\text{aux}}K_{\text{aux}}^\top + \mathbf{1}S^\top)}{\sqrt{C}} V_{\text{aux}} \tag{10}$$

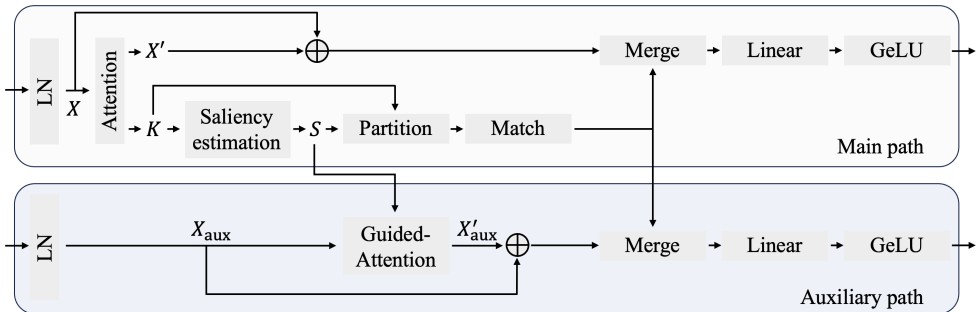

Figure 4: An overview of the learnable video token merging block. The auxiliary path is used during training only.

Table 2: Comparison of the proposed learnable VTM algorithm with conventional algorithms on the LVU dataset.

| Algorithm | Content (↑) | | | Meta data (↑) | | | | User (↓) | | GPU | Throughput |
|---|---|---|---|---|---|---|---|---|---|---|---|
| | Relationship | Speaking | Scene | Director | Genre | Writer | Year | Like | View | | |
| Obj. T4mer (Wu & Krähenbühl, 2021) | 54.76 | 33.17 | 52.94 | 47.66 | 52.74 | 36.30 | 37.76 | 0.30 | 3.68 | - | - |
| VideoBERT (Sun et al., 2019) | 52.80 | 37.90 | 54.90 | 47.30 | 51.90 | 38.50 | 36.10 | 0.32 | 4.46 | - | - |
| Performer (Choromanski et al., 2021) | 50.00 | 38.80 | 60.46 | 58.87 | 49.45 | 48.21 | 41.25 | 0.31 | 3.93 | 5.93GB | - |
| Orthoformer (Patrick et al., 2021) | 50.00 | 38.30 | 66.27 | 55.14 | 55.79 | 47.02 | 43.35 | 0.29 | 3.86 | 5.56GB | - |
| LST (Islam & Bertasius, 2022) | 52.38 | 37.31 | 62.79 | 56.07 | 52.70 | 42.26 | 39.16 | 0.31 | 3.83 | 41.38GB | - |
| ViS4mer (Islam & Bertasius, 2022) | 57.14 | 40.79 | 67.44 | 62.61 | 54.71 | 48.80 | 44.75 | 0.26 | 3.63 | 5.15GB | 25.64 |
| S5 (Wang et al., 2023) | 61.98 | 41.75 | 69.88 | 66.40 | 58.80 | 50.60 | 47.70 | 0.25 | **3.51** | 3.85GB | 25.0 |
| S5+LSMCL (Wang et al., 2023) | 61.98 | 41.75 | 72.53 | 66.40 | **61.34** | 50.60 | 47.70 | 0.24 | **3.51** | 3.85GB | 25.0 |
| Learnable VTM | **64.28** | **42.12** | **75.58** | **70.09** | 59.77 | **53.57** | **48.55** | **0.21** | 4.01 | 1.60GB | 44.94 |

where $\mathbf{1}$ is a $N$ dimensional vector of ones. In the saliency guided attention, the contribution of each token is controlled by its estimated saliency score; if $s_i$ is positive, $i$-th token affects more on $X'_{\text{aux}}$, whereas if $s_i$ is negative it has less influence on $X'_{\text{aux}}$. In other words, it increases the influences of the tokens with high saliency scores in the attention process. Therefore, during training, the network is encouraged to assign high saliency scores to the tokens with meaningful information and low saliency scores to the others to obtain the better predictions. At the first VTM block, the auxiliary path employs the same input with the main path. From the second VTM block, it takes the output of auxiliary path in the previous VTM block as its input.

Also, it is worth pointing out that the auxiliary path is used for the network training only. Compared to other VTM methods, learnable VTM only introduces additional computation of score estimation module, which is fast and light enough, during test. Therefore, it shows almost same inference speed with other VTM methods.

## 4 Experiments

### 4.1 Datasets

**LVU (Wu & Krähenbühl, 2021):** It contains ∼30K videos sampled from ∼3K movies on the MovieClips (mov) website. Most videos are 1 to 3 minutes long. It provides the labels for 9 long-video understanding tasks which can be grouped into three major categories:

- Content understanding: 'relationship,' 'speaking style,' and 'scene/place'
- Metadata prediction: 'director,' 'genre,' 'writer,' and 'movie release year'
- User engagement: 'YouTube like ratio,' and 'YouTube popularity'

As the evaluation metrics, we adopt the top 1 classification accuracy for content understanding and metadata prediction tasks and mean-squared error (MSE) for user engagement tasks.

**Breakfast (Kuehne et al., 2014):** It provides 1,712 videos with the average length of 2.32 minutes and the total length of 77 hours. The videos contain 52 individuals and 18 different backgrounds in total. Each video belongs to one of 10 complex cooking activities.

**COIN (Tang et al., 2019):** It consists of 11,827 videos with the average length of 2.36 minutes, collected from YouTube. Each video belongs to one of 180 distinct procedural tasks.

Table 3: Comparison on the Breakfast dataset. PT stands for pretraining.

| Algorithm | PT Dataset | #PT Samples | Accuracy |
|---|---|---|---|
| VideoGraph (Hussein et al., 2019b) | Kinetics-400 | 306K | 65.50 |
| Timeception (Hussein et al., 2019a) | Kinetics-400 | 136M | 71.30 |
| GHRM (Zhou et al., 2021) | Kinetics-400 | 495K | 75.50 |
| D-sprv (Lin et al., 2022) | HowTo100M | 136M | 89.90 |
| ViS4mer (Islam & Bertasius, 2022) | Kinetics-600 | 495K | 88.17 |
| S5 (Wang et al., 2023) | Kinetics-600 | 495K | 90.14 |
| S5+LSMCL (Wang et al., 2023) | Kinetics-600 | 495K | 90.70 |
| Learnable VTM | Kinetics-600 | 495K | **91.26** |

Table 4: Comparison with the state-of-the-art methods on the COIN dataset. PT stands for pretraining. Here, $*$ means the reproduction results with the official codes.

| Algorithm | PT Dataset | #PT Samples | Accuracy |
|---|---|---|---|
| TSN (Tang et al., 2020) | Kinetics-400 | 306K | 73.40 |
| D-sprv (Lin et al., 2022) | HowTo100M | 136M | 90.00 |
| ViS4mer (Islam & Bertasius, 2022) | Kinetics-600 | 495K | 88.41 |
| ViS4mer$^*$ (Islam & Bertasius, 2022) | Kinetics-600 | 495K | 87.11 |
| S5 (Wang et al., 2023) | Kinetics-600 | 495K | 90.42 |
| S5+LSMCL (Wang et al., 2023) | Kinetics-600 | 495K | **90.81** |
| Learnable VTM | Kinetics-600 | 495K | 88.55 |

## 4.2 Implementation Details

We follow the experimental settings of conventional long-form video understanding algorithms (Islam & Bertasius, 2022; Wang et al., 2023). We employ three transformer blocks in the baseline network. As the encoder, we use ViT-L (Dosovitskiy et al., 2021) pretrained on ImageNet-21K (Ridnik et al., 2021) on the LVU (Wu & Krähenbühl, 2021) dataset and employ Swin-B (Liu et al., 2021) pretrained on Kinetics-600 (Kay et al., 2017) on the Breakfast (Kuehne et al., 2014) and COIN (Tang et al., 2019) datasets. Images are resized to $224 \times 224$ for the feature extraction. Hence, $H = W = 16$ and $C = 1024$ for the LVU dataset and $H = W = 7$ and $C = 1024$ for the Breakfast and COIN datasets. The size of the length of input video for each dataset is also same with (Islam & Bertasius, 2022; Wang et al., 2023): we use 60 frames for the LVU dataset and 64 frames for the Breakfast and COIN datasets. We use the AdamW (Loshchilov & Hutter, 2017) optimizer with a batch size of 16 and a weight decay of 0.01. We set the learning rate to 0.001. We train the network for 70 epochs by using cosine learning rate scheduler (Gotmare et al., 2018) with 10 epochs warm-up. For experiments, we use 8 Tesla V100 GPUs and PyTorch.

## 4.3 Experimental Results

**Comparison on LVU:** In Table 2, we compare the proposed algorithm with the conventional methods on the LVU dataset. With the smallest memory footprint, the proposed algorithm achieves the best scores in 7 out of 9 tasks on the LVU dataset. Performer (Choromanski et al., 2021) and Orthoformer (Patrick et al., 2021) employ the efficient variants of self-attention to reduce the computation costs. The proposed learnable VTM outperforms these approaches with significant performance margins and less GPU memory usages. It shows the efficiency and effectiveness of our approach. Also, ViS4mer (Islam & Bertasius, 2022) and S5 (Wang et al., 2023) adopt S4 layers instead of self-attention layers to capture the long-term dependencies in videos because of its linear computation complexity to the number of input tokens. The promising results of these methods have suggested that S4 layer can be an efficient replacement of self-attention layer for long-form video inputs. However, the higher scores of the proposed algorithm broaden the potential usages of the self-attention layers for various long-form video tasks. Moreover, S5 utilizes LSMCL, which is a pretraining based on the contrastive learning, to boost its performances. Nevertheless, without the time-consuming pretraining, the proposed algorithm yields better scores on the LVU dataset.

Table 5: Comparison of long-video understanding results on the LVU dataset according to $\gamma$.

| Algorithm | Scene | Director | Like | Throughput | GPU |
|---|---|---|---|---|---|
| Baseline | 69.76 | 62.61 | 0.28 | 6.52 | 10GB |
| $\gamma = 2$ | 72.09 | 68.22 | 0.25 | 22.13 | 2.7GB |
| $\gamma = 6$ | 75.58 | 70.09 | 0.21 | 44.94 | 1.6GB |
| $\gamma = 10$ | 74.41 | 70.09 | 0.23 | 48.89 | 1.5GB |

Table 6: Comparison of long-video understanding results on the LVU dataset according to $(L_1, L_2, L_3)$.

| $(L_1, L_2, L_3)$ | Scene | Director | Like | Throughput | GPU |
|---|---|---|---|---|---|
| $(10, 30, 60)$ | 75.58 | 69.15 | 0.21 | 33.62 | 2.7GB |
| $(6, 30, 60)$ | 75.58 | 70.09 | 0.21 | 44.94 | 1.6GB |
| $(4, 20, 60)$ | 73.25 | 68.22 | 0.23 | 53.75 | 1.4GB |

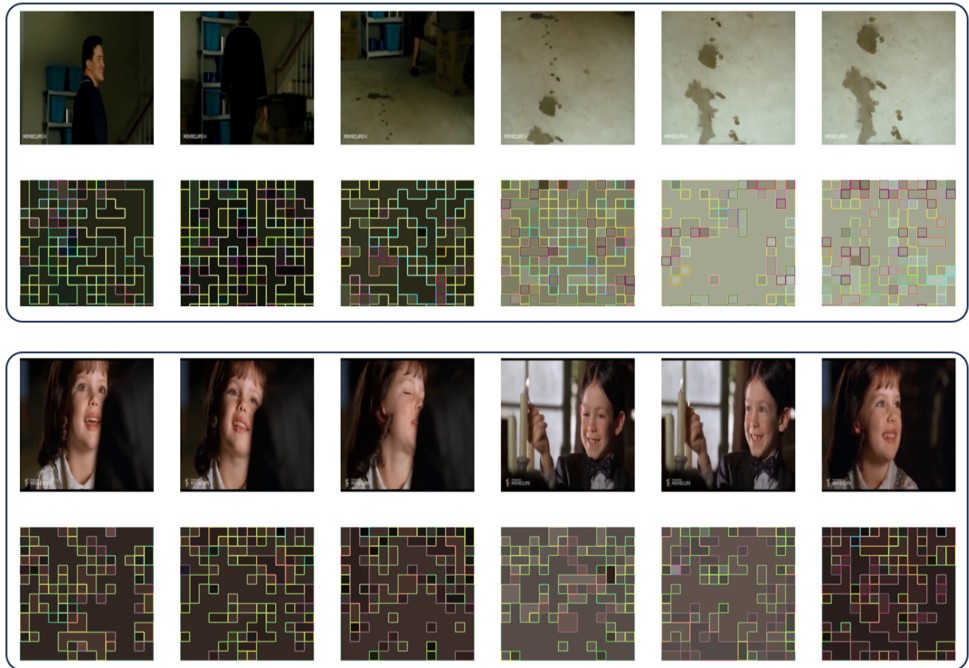

Figure 5: Visualizations of video token merging results on the LVU dataset. Patches with same inner and border color are merged together. The tokens corresponding to the backgrounds are merged together, thereby increasing the influence of salient tokens in the attention process.

**Comparison on Breakfast:** Table 3 compares the performances of the proposed algorithm and the conventional techniques on the Breakfast dataset. The proposed learnable VTM achieves the best score on this challenging long-range activity classification dataset as well. For pretraining, D-sprv (Lin et al., 2022) leverages the HowTo100M (Miech et al., 2019) dataset which contains much more training samples than our pretraining dataset, Kinetics-600 (Carreira et al., 2018). Nevertheless, we outperform D-sprv (Lin et al., 2022) with the accuracy gap of $1.36\%$.

**Comparison on COIN:** Table 4 compares the scores of the proposed algorithm and the conventional techniques on the COIN dataset. We note that the COIN dataset consists of the videos on YouTube and more than 1,000 videos are not available anymore. Therefore, ViS4mer, which is one of the state-of-the-art method on the COIN dataset, achieves only $87.11\%$ accuracy when it is trained on the current version of the COIN dataset. It may be because of many missing training videos. Even though the comparison is not perfectly fair, we report the results on the COIN dataset for reference. The proposed learnable VTM yields better results than ViS4mer with the same training and test data. Also, it shows the comparable score with S5 (Wang et al., 2023).

### 4.4 Analysis

**Analysis on $\gamma$:** Table 5 compares the performances of learnable VTM with different $\gamma$. Compared to the baseline, at all $\gamma$, the proposed algorithm improves the performances. Also, at $\gamma = 10$, it increase the throughput and the memory efficiency by 7.49 and 6.6 times, respectively. At $\gamma = 6$, the proposed algorithm shows the best results.

Table 7: Comparison of long-video understanding results of various VTM designs on the LVU dataset.

| Algorithm | Scene | Director | Like |
|---|---|---|---|
| Weighted average | 72.09 | 68.22 | 0.24 |
| Motion | 74.41 | 64.48 | 0.24 |
| Motion weighted average | 74.41 | 66.35 | 0.23 |
| Learnable | 75.58 | 70.09 | 0.21 |

Table 8: Comparison of throughput and memory footprint of learnable VTM for training and inference.

| Algorithm | Phase | Throughput | Memory |
|---|---|---|---|
| ViS4mer (Islam & Bertasius, 2022) | Inference | 25.64 | 5.15GB |
| S5 (Wang et al., 2023) | Inference | 25 | 3.85GB |
| Learnable VTM | Training | 27.84 | 2.8GB |
| Learnable VTM | Inference | 44.94 | 1.6GB |

**Analysis on** $(L_1, L_2, L_3)$**:** Table 6 shows the results of the proposed learnable VTM with different $(L_1, L_2, L_3)$. Note that $i$-th VTM block takes the tokens in $L_i$ consecutive frames as its input at a time. At $(L_1, L_2, L_3) = (4, 20, 60)$, the proposed algorithm yields the lowest scores, since it can not capture the long temporal dependency in the early stage of the network. We see that the proposed algorithm yields the best scores at $(L_1, L_2, L_3) = (6, 30, 60)$.

**Weighted average pooling:** To merge tokens, we use average pooling as the default setting in all VTM methods. However, once tokens are merged, they may represent more than one input patch. Thus, to reflect the token size in merging process, we combine tokens by averaging weighted by their sizes. However, as shown in Table 7, this weighted average pooling decrease the performances of learnable VTM. Thus, we exploit the average pooling to merge tokens.

**Motion weighted average pooling:** In motion-based VTM, we can combine tokens by averaging weighted by their motion magnitudes. Table 7 shows the performances of motion-based VTM with the motion weighted average pooling. It yields the similar scores with the standard motion-based VTM with the average pooling.

**Complexity:** Table 8 compares the throughput and memory footprint of learnable VTM during training and inference. Since the auxiliary path is additionally employed during training, it requires more computation costs. However, even during training, learnable VTM is still faster than conventional methods including ViS4mer (Islam & Bertasius, 2022) and S5 (Wang et al., 2023) and it also requires less amounts of memory than them.

**Visualizations:** In Figure 7, we visualize the tokens merging results at the end of the network over multiple frames of video. We see that tokens with similar semantics are merged together. Also, tokens corresponding to backgrounds or unnecessary informations are merged more than tokens corresponding to salient objects. It is because learnable VTM selects tokens with high saliency scores as the target tokens. More visualization results are available in the supplemental document.

## 5   Conclusion

In this paper, we investigate the video token merging techniques for long-form video data. Unlike previous algorithms that apply uniform partitioning and merge tokens solely based on the visual similarity, we argue tokens with different saliencies should be treated unequally to avoid undesirable information loss after merging important tokens. To this end, we explore various video token merging methods and receive interesting intuitions from region-concentrated and motion-based token merging results. Lastly, we propose a learnable video token merging scheme that adaptively samples target tokens and learns discriminative representations from the long-form videos. Compared to the baseline, the proposed algorithm achieves substantial improvements in terms of the performance, memory cost and throughput.

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

# A  More Implementation Detail

## A.1  Network Architecture

Figure 6 illustrates the detailed architecture of the proposed learnable VTM. For all datasets, the encoder extracts the tokens with 1024 channel dimension. We note that the linear layer in each VTM block reduces the channel dimension into half of it. Hence, after third VTM block, each token has 256 channel dimension. Also, the auxiliary path is only used for network training.

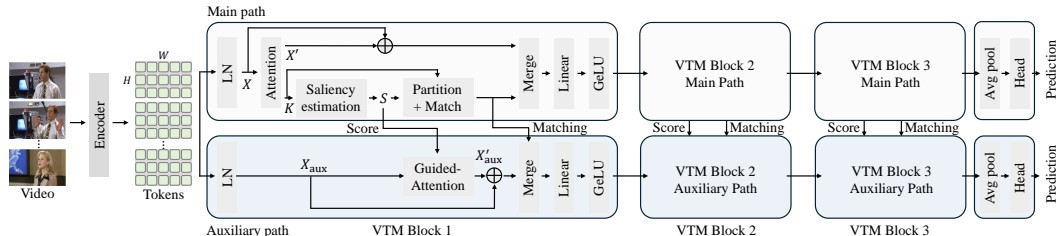

Figure 6: A network architecture of the proposed learnable VTM.

# B  More Experiments

## B.1  Analysis on $R$

Table 9 compares the results of the proposed algorithm at different $R$. Note that $R$ denotes the number of merged tokens at each VTM block. At $R = |\mathcal{S}|$, we merge all source tokens with target tokens. However, there may exist some source tokens which does not have target tokens with similar semantics. Thus, at $R = |\mathcal{S}|$, undesirable merging of tokens can happen, thereby decreasing the performances. On the other hand, at $R = 0.5|\mathcal{S}|$, only the half of source tokens are merged with target tokens, and thus some source tokens may not be merged even though they have similar target tokens. Therefore, the proposed algorithm shows the best scores at $R = 0.8|\mathcal{S}|$.

Table 9: Comparison of long-video understanding results on the LVU dataset according to $R$.

| $R$ | Scene | Director | Like |
|---|---|---|---|
| $|\mathcal{S}|$ | 74.41 | 66.40 | 0.22 |
| $0.8|\mathcal{S}|$ | 75.58 | 70.09 | 0.21 |
| $0.5|\mathcal{S}|$ | 72.09 | 67.28 | 0.22 |

# C More Visualizations

Figure 7 visualizes the token merging results of the proposed learnable VTM on the LVU dataset.

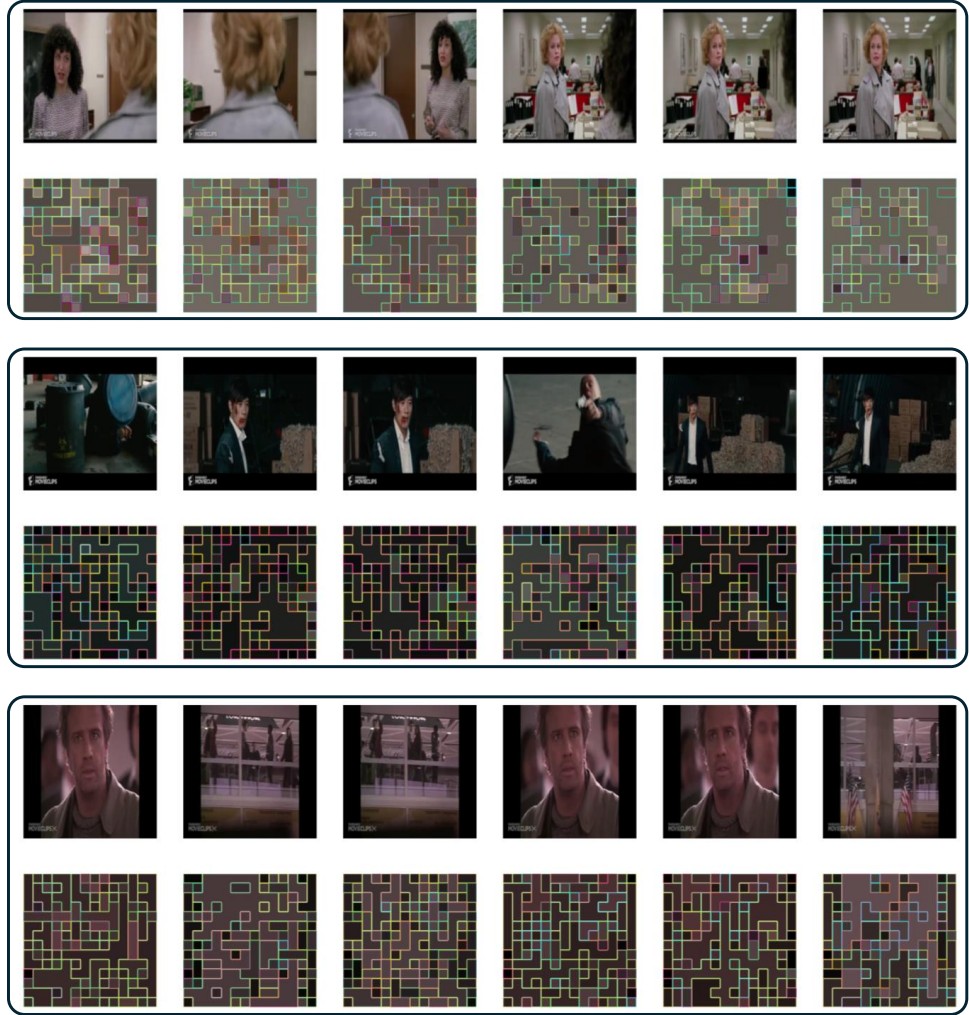

Figure 7: Visualizations of video token merging results on the LVU dataset. Patches with same inner and border color are merged together.

