# OpenReview forum: "Video Token Merging for Long Video Understanding"
_NeurIPS.cc/2024/Conference — NeurIPS 2024 poster_

### Official Review · Reviewer_JTuZ · 2024-07-10

**Soundness:** 3
**Presentation:** 3
**Contribution:** 2
**Rating:** 5
**Confidence:** 3

**Summary:**

- This paper carries out an analysis of token merging[1] in the context of long-form video understanding and proposes learnable video token merging (VTM) to select semantics/saliency guided tokens for merging.
- In token merging, at each layer, the tokens are divided into two sets source S and target T through uniform sampling. Tokens in S are matched to T based on similarity and merged (usually by average pooling). This paper compares this naive VTM with two other variants where selection of T is guided by informed heuristics: (1) region VTM where tokens at the center of each frame are more likely to be retained, (2) motion VTM where tokens with high motion are more likely to be retained. Through this analysis, authors argue that the strategy to select T plays an important role in the final performance.
- Motivated by this, authors propose a learnable VTM where it first predicts a saliency score for each input token. The target set T is sampled according to the probability distribution defined by saliency score. Since this partition operation is not differentiable, authors propose a novel training architecture where a parallel auxiliary network is trained alongside. The saliency scores are used to bias the attention score of the aux network, thereby supervising the saliency prediction to focus on important tokens. Aux network can be discarded at test time.
- Authors carry out a fair evaluation of the learnable VTM on LVU, Breakfast and COIN datasets by comparing against several baselines including ViS4mer, S5, D-sprv. Learnable VTM performs better than baselines in almost all evaluation tasks with low GPU memory usage and high throughput.

[1] Daniel Bolya, Cheng-Yang Fu, Xiaoliang Dai, Peizhao Zhang, Christoph Feichtenhofer, and Judy Hoffman. Token Merging: Your ViT but faster. In ICLR, 2022.

**Strengths:**

- In learnable VTM, the idea of learnable and auxiliary path is interesting. There is no way to directly supervise the token saliency prediction of the main path because partition operation is non-differentiable. Hence the attention in auxiliary path is influenced by saliency scores of the main path, which encourages the saliency prediction to focus on important tokens.
- The evaluation is fair and consistent. The authors use the same backbone as the prior works to encode input video frames, thereby ensuring a fair evaluation.
- The results of learnable VTM on LVU dataset and Breakfast is noticeably better than baselines with less GPU memory usage. However, on COIN dataset, it doesn't perform better than S5 baseline.

**Weaknesses:**

### Major weaknesses
- One of the cited contribution is the exploration of region based and motion based VTM (Section 3.3) but it seems trivial. The effectiveness of token selection is already shown in learnable VTM. In light of that, there is an unreasonable focus section 3.3 which is unnecessary.
- Section 3.4 explains little about the details of learnable VTM, how it is trained, how the gradients flow in the presence of non-differentiable partition function, etc.
- There are some stretched claims based on qualitative and quantitative results. For example,
  - In Line 174, authors claim that center VTM performs better than naive VTM. However, according to Table 1, the results are mixed at best.
  - In Fig 5, authors also claim that the visualization of merged tokens show saliency based merging. However, the figure doesn't support the claim. There are many merged tokens on important salient features and some background tokens are not merged.

### Minor issues
- Line 19: it should be "into the domain of video computer vision" as all cited papers are video learning papers.
- Is there a difference between notation of C and D? It looks like both are used interchangably to denote token dimension.
- Table 2: How it throughput measured? fps?

**Questions:**

- How is learnable VTM trained? From Fig 4, it looks like there are two outputs from the network. Do you apply the same loss on both outputs?
- In Fig 4, what does the 'Merge' operation in auxiliary path do? Does it mean that the main and aux - both paths use the same target set sampled by the partition-match of the main path?

---

> ### Author Rebuttal · Authors · 2024-08-07
>
> Thank you for your positive and valuable comments. Please find our responses below.
>
> ***
>
> > **Exploration**
>
> Compared to image token merging methods, token merging for video is relatively under-researched. In this work, we investigate various video token merging methods and finally propose a learnable VTM that outperforms all previous techniques. Like our algorithm, there are many conventional algorithms in the video computer vision community that proposed simple but effective algorithms, such as 3D-CNN, inflated 3D kernel, (2+1)D CNN and etc. Hence, we believe that the exploration of various basic token merging methods has its benefits for the research community and the proposed learnable VTM algorithm will serve as a strong baseline. However, as the reviewer suggested, we will reduce Section 3.3 and include more analysis on learnable VTM.
>
> > **Training of learnable VTM**
>
> Learnable VTM is optimized to reduce the classification losses on both main path and auxiliary path. Also, the entire process is differentiable because we merge the matched tokens by using average pooling as in the standard token merging. Partitioning and matching processes do not have to be differentiable for end-to-end backpropagation. These processes do not update the features but just determine which tokens to merge. We will include this explanation in the revision.
>
> > **Stretched claims**
>
> We agree with the reviewer. We will revise the draft to explain the experimental results precisely.
>
> > **Citations**
>
> We will revise the draft as suggested.
>
> > **Notation C and D**
>
> We will revise the notation more clearly.
>
> > **Throughput**
>
> Yes, we measure the FPS.
>
> > **Training**
>
> In learnable VTM, we apply the same video classification loss to both predictions from the main path and the auxiliary path. Therefore, the saliency estimation module is optimized to assign high saliency scores to the important tokens to reduce the classification loss during training, since the tokens with high saliency scores have more influence in the guided attention in the auxiliary path.
>
> > **Figure 4**
>
> Yes, tokens in the auxiliary path are also merged according to the matching results from the main path.
>
> ***
> We will address all these comments faithfully in the final paper. If you have additional comments, please let us know. Thank you again.

---

> > ### Comment · Reviewer_JTuZ · 2024-08-13
> > **Post-rebuttal comment**
> >
> > Thanks for a detailed rebuttal.
> >
> > ### Major comments
> > - It looks like learnable VTM is very similar to other token sampling method ('Adaptive Token Sampling For Efficient Vision Transformers'). However, such prior token sampling method is parameter free and can be used on any pretrained transformers. Learnable version of such sampling is perceiver module ('Perceiver IO: A General Architecture for Structured Inputs & Outputs'). Authors haven't compared with either of them. Can the authors provide more comments on this?
> > - The visualization of saliency in Fig. 3 doesn't support the claim that the salient patches are being selected. Rather, it looks like that the network is repurposing the unimportant tokens to store information (this has been explored in 'Vision transformers need registers'). This also reaffirms by belief that Learnable VTM may be just a resampler and hence doesn't actually learn vision saliency.
> >
> > ### Minor comments
> > - I agree that the exploration is on different token merging techniques is valueable in the supplementary material. However, it would be better if its length in main paper is reduced.

---

> > > ### Author Response · Authors · 2024-08-13
> > >
> > > Thank you for your additional feedback. Please find our follow-up response below.
> > > ***
> > >
> > > > **Comparison with token reduction methods**
> > >
> > > Token merging and token sampling/dropping are two alternatives to reduce the valid number of visual tokens in the transformers, which reduce the computational cost of the self-attention layer. However, the scope of adaptive token sampling and its similar works are different from our work. They focus on improving the efficiency of the transformer while maintaining similar performance. In this paper, we argue that these methods may not be favorable in the long-form video understanding. Images as well as short-form videos are easy to build short-term dependencies within one image or highly overlapped video frames from a few seconds long, e.g. the basketball and basketball court, the sky and cloud. Thus, a parameter-free or a light weight module can produce reasonably good results for merging and sampling. On the other hand, in long-form videos, it is more challenging to capture dependencies between sparsely sampled frames, e.g. leveraging dependencies to predict the genre and production year of a movie.
> > > As this work specifically focuses on the token merging method, we only include previous merging ideas in the comparison. We believe it is a great idea to expand the scope of this work in the future including adaptive token sampling and Perceiver as two representatives in the concept of general token reduction analysis.
> > >
> > > > **Figure 3**
> > >
> > > Also, as stated in L206-208, we sample the target tokens based on the probability which is computed from the estimated saliency scores. Hence, it is more likely that the tokens with high saliency scores are selected as target tokens, but not always. Therefore, in Figure 3, some tokens corresponding to backgrounds are selected as target tokens. However, we can see that target tokens are selected more around salient objects (two main characters). Moreover, it is desirable that some tokens from backgrounds are selected as target tokens so that source tokens corresponding to background can be merged with those similar target tokens. Otherwise, source tokens from background areas would be merged with unsimilar target tokens corresponding to salient objects, which causes significant loss of important information.
> > >
> > > > **Paper organization**
> > >
> > > Thanks for your advice. As suggested, we will move the explanation on different token merging methods to Appendix and use the saved space to address the comments from all reviewers.
> > >
> > > ***
> > > Thank you again for your time and effort for reviewing our paper. We do appreciate it. If you have additional comments, please let us know.

---

### Official Review · Reviewer_7bwU · 2024-07-10

**Soundness:** 3
**Presentation:** 3
**Contribution:** 3
**Rating:** 4
**Confidence:** 3

**Summary:**

This paper explores various video token merging strategies in the context of long-form video classification and finally propose a learnable Video Token Merging algorithm that dynamically merges video tokens based on visual salient areas. The contributions are summarized as follow:
1.  Explore various video token merging methods including the naïve VTM, the region concentrated VTM, and the motion-based VTM.
2.  Propose the learnable video token merging algorithm, which estimates the saliency scores of each token and adaptively merge visual tokens based on those scores.
3. The proposed algorithm achieves the best or competitive results on various datasets.

**Strengths:**

1. This paper explores various video token merging methods including the naïve VTM, the region concentrated VTM, and the motion-based VTM.
2. Compare with baseline and rule-based video token merging. The proposed learnable video token merging strategy has large improvement.
3. The two-paths design to deal with non-differentiable problem in partitioning process is interesting.

**Weaknesses:**

1. This paper proposes a leanable video token merging strategy. The similiar high-level idea can be found by CTS[1] in image domain. The novelty is insufficient。
2. This paper focuses on video token merging. However, I do not observe any specific design tailored for the video domain in terms of the methodology. let alone long video.



[1] Content-aware Token Sharing for Efficient Semantic Segmentation with Vision Transformers

**Questions:**

1. Due to the two-paths design, has the training time doubled?
2. The paper tries to learn saliency scores using matrix $U_s$. How about using $\sum{QK^T}$ in Equation 8 as saliency scores for each visual token?

**Limitations:**

Yes

---

> ### Author Rebuttal · Authors · 2024-08-07
>
> We do appreciate your constructive comments and will address them faithfully in the final paper. Please find our responses below.
>
> ***
>
> > **Difference from CTS**
>
> CTS is not similar to the proposed algorithm, since it is not even a token merging method. CTS is a semantic segmentation algorithm which shares some neighboring tokens expected to belong to the same class in the segmentation network. Hence, CTS does not have any similar key idea such as learning the token saliency or merging tokens. We will do our best to answer your concerns if you provide more detailed points.
>
> > **Design for video domain**
>
> Compared to the images, there is much more redundant or noisy information in the videos, which deteriorates the quality of video understanding. The proposed learnable VTM is a domain-specific algorithm for long-context scenario. To reveal the discriminative information among redundant tokens in the video, we develop the learnable VTM which learns the saliency of each token and keeps the salient tokens, thereby increasing the influence of these tokens in the following attention processes after merging. Also, we explored the motion-based VTM, which selects the tokens with large motions as the target tokens, and thus this token merging method is only applicable to the video.
>
>
> > **Training time**
>
> Below, we compare the training time. The training time is slightly increased, but not doubled. We use 8 Tesla V100 GPUs and PyTorch for the experiments as stated in L258.
>
> | Naive VTM | Learnable VTM |
> |-----------|---------------|
> | 4.0h      | 3.7h          |
>
>
> > **Attention as saliency score**
>
> The objective of saliency estimation is to assign high scores to the tokens corresponding to important objects and low scores to them corresponding to backgrounds or noisy information. However, $\sum QK^t$ would assign high scores to the tokens which have many similar tokens regardless of their semantics. Therefore, we employ the saliency estimation module for the proposed learnable VTM.
>
> ***
> If you have any additional concerns, please let us know. We will address them faithfully. Thank you again for your constructive comments.

---

> > ### Author Response · Authors · 2024-08-11
> >
> > There was a minor error in above Table for training time. The correct table is as below. We do apologize for the confusion.
> >
> > | Naive VTM | Learnable VTM |
> > |-----------|---------------|
> > | 3.7h      | 4.0h          |
> >
> > ***
> > Thank you again for your time and effort for reviewing our paper. We do appreciate it.

---

> ### Comment · Reviewer_7bwU · 2024-08-13
>
> 1. Design for video domain
>
> The author, although exploring motion-based VTM, ultimately adopts a learnable VTM. In the learnable VTM, I do not see any special design for the video.
>
> 2. Training time
>
> Could you explain why maintaining two paths during training does not result in a relationship of twice as much?

---

> > ### Author Response · Authors · 2024-08-13
> >
> > Thank you for your constructive review and insightful suggestions, all of which will be addressed faithfully in the final paper. Please find our responses below.
> > ***
> >
> > > **Design for video domain**
> >
> > We kindly request Reviewer 7bwU to clarify the definition of ‘special design for video’. In recent transformer-based networks and many foundation models, various inputs, including images, short-form videos, long-form videos, and text, are all processed in the form of tokens, but they just have different modalities and properties. If the absence of special design for video means that our method can be ‘generally’ applied to different modality tokens without raising errors, it is true. However, as stated in Section 1 and our previous response, the proposed algorithm is designed to handle long-contextual information in the video more efficiently and effectively. Compared to relatively well-structured text inputs and images/short-videos with relatively small amounts of information, it is more challenging to capture short-term and long-term dependencies across tokens in the long video since the long video contains many tokens which convey complex semantics individually or collectively. Therefore, we design our algorithm by merging tokens adaptively based on saliency scores for capturing the token dependency more easily while minimizing the information loss. Also, please note that the proposed algorithm achieves better results than all conventional algorithms for long-video understanding on various datasets. We believe that it indicates that the proposed algorithm has efficient design for long-video understanding.
> >
> >
> > > **Training Time**
> >
> > We compare the throughput (FPS) in Table below. The throughput of learnable VTM with the auxiliary path (during training) is almost half, compared to that of naive VTM or that of learnable VTM without the auxiliary path (during inference). This is also shown in Table 9 in Appendix B.2. However, when measuring training time, there are various different factors such as data preprocessing, data loading, logging, and back-propagation. These factors take much longer than the forward path, and thus the total training time of learnable VTM is not doubled.
> >
> > |            | Naive VTM | Learnable VTM (training) | Learnable VTM (inference) |
> > |------------|-----------|--------------------------|---------------------------|
> > | Throughput | 45.39     | 27.84                    | 44.94                     |
> >
> >
> >
> > ***
> > If you have any additional concerns, please let us know. Thank you again for your comments.

---

### Official Review · Reviewer_sUxH · 2024-07-13

**Soundness:** 4
**Presentation:** 4
**Contribution:** 3
**Rating:** 6
**Confidence:** 4

**Summary:**

The paper approaches the task of long-video understanding from token reduction perspective. Specifically, Transformer-based approaches suffers from memory bottleneck and quadratic computation complexity with increasing number of tokens, which is even more pressing with long-videos as input. The paper builds on a recently developed token merging mechanism, and proposes a learned saliency measure to modulate what tokens gets merged instead of using a random or hand-crafted saliency measure. The central hypothesis of the work is that typically techniques that use similarity as merging criteria may inadvertently lose out on salient tokens. The paper reports experiments on three conventional long-video benchmarks (LVU, Breakfast and COIN), and shows effectiveness of their approach compared to prior related works both in terms of performance and memory requirement. The paper also ablates the effectiveness of their proposed saliency measure (learned VTM) over hand-crafted measures including motion-based (using optical flow), center-biased and random schemes.

**Strengths:**

- The paper is well-written with most of the information presented for ease of understanding
- The memory requirement is lower than S4, with competitive performance which highlights the importance of token selection in the case of long-videos

**Weaknesses:**

- Comparison to related token saliency approaches
    - The paper proposes a scheme to identify salient tokens by using a learned projection matrix $U_s \in \mathcal{R}^{D \times 1}$ with $\texttt{tanh}$ activation function
    - However, learnable token saliency methods have also been used in prior works, such as EVEREST [1], which uses a pair-wise learned saliency at feature-level (equation 2) using $\texttt{Conv3d}$. The resulting approach was shown to be effective in the Masked Autoencoding setup
    - Having a motion-based merging scheme is a good baseline, but some variants of learnable token saliency could also be tried to gain better understanding how token saliency gets influenced by different approaches

- Role of $L_1, L_2, L_3$
    - The paper proposes to take tokens from $L_i$ consecutive frames for the $i^{th}$ VTM block
    - It seems that choosing the values of $L_i$ is quite crucial given its impact on performance and memory requirement (Table 6) that forms the central claim of the paper
    - However, the paper highlighted the contribution of token saliency more compared to the choice of $L_i$ hyperparameters
    - Did the authors experiment with a rather simplistic setup using a single VTM block and/or with all $L_i$ being 60? It would help the readers to gain better understanding of what works in long-videos


- How saliency changes with tokens from different number of frames?
    - It seems that the saliency is being computed at each VTM block. It would be interesting to see how the saliency changes across the three VTM blocks
    - On that note, what VTM block’s saliency is being visualized in Figure 5?



### Minor
- Line 145-146: “$i$-th transformer block takes the tokens corresponding to $L_i$ frames without overlapping”
    - Confusing when $i$ is referred to as the frame number and the block number of transformer at the same time
- Line 145-147: $j$ is not defined


### Typos
- Line 24-25: “so the tokenizing the”
- Line 36: “selectio”
- Line 114-115: “applications are mostly remained”
- Line 117: “depedencies”
- Line 161: “in the videos, .”
- Line 165: “regarding less of the”
- Line 177: “sailent”, “the the”
- Line 306: “sailencies”


### References
[1] “EVEREST: Efficient Masked Video Autoencoder by Removing Redundant Spatiotemporal Tokens”. Sunil Hwang and Jaehong Yoon and Youngwan Lee and Sung Ju Hwang. ICML 2024.

**Questions:**

- Line 141: why L >= 60?
- Is saliency projection used in all auxiliary VTM blocks?

---

> ### Author Rebuttal · Authors · 2024-08-07
>
> Thank you for your positive review and insightful suggestions, all of which will be addressed faithfully in the final paper. Please find our responses below.
>
> ***
>
> > **Difference from EVEREST**
>
> The proposed learnable VTM is quite different from EVEREST. EVEREST measures the cosine similarity of tokens at the same spatial location in consecutive frames for more efficient mask generation which is used for masked video autoencoder training. Therefore, by its design, it does not consider the content of each token for saliency prediction. In contrast, learnable VTM employs the auxiliary path and the saliency guided attention to learn the saliency of each token. Therefore, the saliency estimation module is optimized to assign the high saliency scores to important tokens to reduce the classification loss in the auxiliary path.
>
> > **Learnable VTM variants**
>
> Below, we compare the variants of learnable VTM. In 'From $X$’, we estimate the saliency scores from token features $X$ instead of key vectors $K$. In `Multiply’, we use $Q_\mathrm{aux}K_\mathrm{aux}^t\odot\mathbf{1}S^t$ inside Softmax in Eq.(10) for the saliency guided attention. Here, $\odot$ denotes the Hadamard product. Both alternatives show decent scores. However, the proposed learnable VTM yields better performances and thus we employ it as our method.
>
> | Methods       | Relationship | Scene | Director | Writer |
> |---------------|--------------|-------|----------|--------|
> | From $X$      | 59.52        | 75.58 | 63.55    | 51.19  |
> | Multiply      | 59.52        | 72.09 | 70.09    | 46.42  |
> | Learnable VTM | 64.28        | 75.58 | 70.09    | 53.57  |
>
>
>
> > **Impact of $L$**
>
> Yes, as the reviewer pointed out, $L_i$ is an important hyper-parameter for both efficiency and effectiveness of the proposed algorithm. Therefore, we analyze it in Table 6. However, the performance gain of learnable VTM is not just from the selection of $L$, because the other VTM methods in Table 1 yield low scores at the same $L$ setting. We will discuss this in the revision.
>
> > **$L>60$**
>
> We evaluate the proposed algorithm with $L=60$ since conventional algorithms use 60 frames for their evaluation. Below, we list the performances of learnable VTM with $L=100$ setting. We see that the proposed algorithm yields good results in this setting as well.
>
> | Methods | Relationship | Speaking | Scene | Director | Genre | Writer | Year  | Like | View |
> |---------|--------------|----------|-------|----------|-------|--------|-------|------|------|
> | $L=60$  | 64.28        | 42.12    | 75.58 | 70.09    | 59.77 | 53.57  | 48.55 | 0.21 | 4.01 |
> | $L=100$ | 67.11        | 42.12    | 75.58 | 68.22    | 61.34 | 52.38  | 48.55 | 0.22 | 3.85 |
>
>
>
> > **Saliency visualization**
>
> We will include the visualization results in the final copy.
>
> > **Figure 5**
>
> In Figure 5, we visualize the token merging results at the last VTM block, as stated in L298-299. We will revise this more clearly in the revision.
>
> > **L145-147 and typos**
>
> We will revise them thoroughly. Thank you for your suggestion.
>
>
>
> ***
>
> If you have any additional concerns, please let us know. We will address them faithfully. Thank you again for your positive comments.

---

### Official Review · Reviewer_rFDe · 2024-07-13

**Soundness:** 3
**Presentation:** 2
**Contribution:** 2
**Rating:** 5
**Confidence:** 4

**Summary:**

This paper builds on Token Merging (ToMe) to improve its performance. In particular, the authors explore different ways to partition tokens so that the merging operation can lead to better performance while maintaining speed. They explore region-concentrated merging, motion-vector based merging and a learnable selector, and find that the learnable version works best. To make the network trainable, they employ a creative auxiliary path mechanism  to make everything differentiable. They find that their learnable VTM obtains good results compared to baselines on long form datasets (LVU, Coin, Breakfast), and that it outperforms the other methods they introduce.

**Strengths:**

The problem this paper addresses is an important one. Videos (especially long ones) have many redundant tokens and reducing their number while maintaining performance is a crucial problem to solve in the field.

The model itself is well designed and uses a creative auxiliary path to handle a non-differentiable partitioning process. Given the premise of the paper, the model is well-designed and seems to address the issue they propose.

I also appreciate the exploration of different methods, and a comparison on which worked better. This kind of analysis is often missing from papers and I am grateful for the authors for including it.

**Weaknesses:**

I don’t really agree with the premise of the paper (and am open to a rebuttal to explain if I’m wrong here). Token Merging already explored merging for video in detail. The reason Token Merging is based on similarity is that by combining tokens that are extremely similar, the weighted average of the those tokens should produce an extremely similar result in the attention operation. This was also detailed more in the Token Merging follow-up TomeSD. If you use different criteria such as saliency (which is not really well-defined), this is no longer guaranteed, and from equation (10) it seems like the authors do not use the proportional attention scheme from ToMe (Eq 1 in the original paper). Table 8 doesn’t show the learnable method using this; it seems to just be about the pooling part rather than the attention operation.

I also don’t understand the intuition behind the saliency: shouldn’t we be aiming to combine together tokens that are NOT relevant, so that the transformer can focus more on the relevant tokens, rather than averaging (and thus losing) information from the more salient / important tokens? I’d really appreciate some clarification here. From Figure 3, it doesn’t look like learnable VTM is focusing on visually important tokens: it’s picking ones from the ceiling and wall in addition to the people.

My main issue is with the evaluation. The evaluation seems not quite fair, especially when measuring memory usage and throughput. Shouldn’t it be compared to baseline merging algorithms, like the naive ToMe? My impression is that the memory usage and throughput from VTM will be exactly the same as ToMe because it uses a similar partitioning scheme and constant factor reduction, which is why it may not be included in the results, but this seems important to include for context. Furthermore, the improvement on metrics is quite small, given that the speed is the same as other merging methods. Is this expected?

Also, The paper is motivated by “long-term” video, but evaluates on 64 frames, which isn’t really long and in my view, doesn’t merit only evaluating on LVU, Breakfast and COIN. Kinetics-400 has 300 frames per video, and is a more standard benchmark for evaluating video backbones - in fact, the original ToMe paper includes experiments on those datasets, which would make for a more fair comparison. Furthermore, nothing about the method itself is specific to these longer videos. I think evaluating on more standard datasets is crucial to measuring the actual strength of the method, especially compared to baselines like Token Merging. In particular, the long-form datasets are very compressible.

The paper is not well-written and the grammar needs a lot of revision, making it hard to focus on the content of the paper itself. In addition, a lot of space is spent on methods that are not really used in the final results (center, region, motion vector) and on citing equations from preliminary works (token merging, attention). Given that a claimed contribution is an exploration of these different methods, I would also have expected more detailed ablations and experiments to understand exactly why some of the methods perform better than others.

**Questions:**

It’s not really expected that VTM (or any merging method) should score better than full attention, as it has strictly less information. This is backed up in the Token Merging and other follow up papers. Why is it expected (as said on L160) that merging should perform better? It’s supposed to be just faster, with a minimal drop in performance.

The motion vector method requires extracting pre-computed motion vectors from the encoded video. However, those are computed for 30 FPS, and for the original video size, meaning they don’t actually apply to downsampled (64 frames, 224x224). Was this taken into account? It’s certainly not fast to re-compute these motion vectors if you’re doing random cropping or frame selection.

Is it possible to know the effect on training wall-clock time from this method? This the the metric that practitioners really care about, so including this would potentially strengthen the results of the paper.

**Limitations:**

The biggest limitation of this compared to baseline Token Merging is that it requires re-training. ToMe could be applied to a pre-trained network out-of-the-box. Learnable VTM cannot do this, making it impossible to apply to a pre-trained video network. The other proposed methods (region, motion vector) can do this though, and this would be a good thing to note somewhere in the paper.

In my view, the limitation of methods like VTM is that it always reduces a constant number of tokens per video, even though some videos are inherently more compressible than others. For example, Breakfast videos are extremely compressible compared to the average LVU video. However, this is beyond the scope of the paper, and using constant reduction is certainly more convenient, but this would be a good limitation to acknowledge and perhaps address in the future.

---

> ### Author Rebuttal · Authors · 2024-08-07
>
> We do appreciate your constructive comments and will address them faithfully in the final paper. Please find our responses below.
>
> ***
>
> > **Token merging for video**
>
> To the best of our knowledge, there are only a few techniques for video token merging, such as TESTA(EMNLP2023), ToMe(ICLR2023), and VidToMe(CVPR2024). However, all of them are straightforward applications of the standard token merging to video input. Specifically, TESTA and VidToMe employ the same token partitioning, matching, and merging scheme of ToMe.
> In contrast, we explored multiple token merging methods for video and proposed the learnable token merging method which shows good performances on various long-video classification datasets.
>
>
> > **Saliency**
>
> Learnable VTM uses the estimated saliency scores to divide the tokens into target token set and source token set. The saliency score estimation aims to select a greater number of important tokens (e.g. tokens corresponding to important objects in the scene) as target tokens, thereby increasing the influence of these tokens in the following attention processes after merging.
>
> As in the standard token merging, we merge tokens based on their similarity. Hence, even though some tokens corresponding to the important objects are selected as target tokens, they are not merged with irrelevant tokens, and thus we do not lose much important information during the merging process. It is also shown in Figure 5. We can see that tokens are merged more in the background area than in the salient area.
>
> Also, as stated in L206-208, we sample the target tokens based on the probability which is computed from the estimated saliency scores. Hence, it is less likely that the tokens with low saliency scores are selected as target tokens, but not always. As shown in Figure 3, some tokens corresponding to backgrounds or unimportant objects can be selected as target tokens.
>
> > **Comparison**
>
> Please note that ToMe (ICLR2023) does not provide the results on the long video classification datasets. Kinetics only includes 10s video clip which is spatially heavy and without many temporal dependencies. From previous literature, it is easy to receive good performance with a single frame (What Makes a Video a Video: Analyzing Temporal Information in Video Understanding Models and Datasets, CVPR 2018), which is very different from the scope of this paper.  As stated in L154-155, our naive VTM is a straightforward application of ToMe which exploits the standard token merging method as intact as possible. As the reviewer pointed out, the memory usage and throughput of various VTM methods that we explored in this paper are almost the same. However, with the same memory and speed, a learnable VTM shows better performances than other VTM methods. Moreover, the proposed algorithm outperforms the conventional long-video understanding algorithms with large margins even though it has much better memory-efficiency and speed.
>
> > **Long video understanding**
>
> For evaluation, we just follow the standard protocol in the field of long video understanding. For fair comparison, we use the same datasets (LVU, COIN, and Breakfast) and the same input setting as in conventional algorithms (ViS4mer, S5) for long video understanding.
>
>
> > **High performance of VTM**
>
> In the long video understanding datasets such as LVU, there is a lot of redundant and noisy information in a video. Therefore, as shown in Table 1, the transformer network does not yield reliable prediction results because those redundant tokens may hinder the feature refinement in the attention layers. However, when tokens are merged properly, the influence of important tokens in the attention process increases and thus the network can yield better performances, as shown in our experimental results. This phenomenon has also been revealed in the recently published token selection work (Selective Structured State-Spaces for Long-Form Video Understanding, CVPR 2023).
>
> > **Motion vector computation**
>
> As stated in L194-195, motion decoding takes only 0.3 milliseconds for each frame which is negligible, since the motion is converted to the token resolution $H \times W$ by using average pooling.
>
> > **Training time**
>
> We list the training time for the LVU and COIN datasets below. The training is finished within a few hours. We use 8 Tesla V100 GPUs and PyTorch for the experiments as stated in L258.
>
> | Methods       | LVU | COIN |
> |---------------|-----|------|
> | Learnable VTM | 4.0h | 3.5h  |
>
>
>
> > **Re-training, fixed compression ratio**
>
> We agree with the reviewer. We will include this discussion in the revision.
>
>
> ***
>
> If you have any additional concerns, please let us know. We will address them faithfully. Thank you again for your constructive comments.

---

> ### Comment · Reviewer_rFDe · 2024-08-10
> **Raising Rating + Followup response**
>
> Thanks for the detailed rebuttal, I really appreciate your efforts. I will raise my rating for now to a 4. I have some further doubts that I'd appreciate clarification on.
>
> __Saliency__
>
> I think I understand this better now but I really don't think saliency is the right word. From what I understand, the point of this is to learn some scoring function to do a better job of partitioning, since the hypothesis of the paper is that the partitioning scheme of ToMe is suboptimal. Maybe "learned partition" coudl be a better word? Saliency is too broad / vague of a term.
>
> __Training Time__
>
> When I asked for the training time, I meant compared to standard ToMe (no learned module) and a baseline with no merging. From the results of the original ToMe paper they demonstrated very large speedups on training time. It seems that learnable VTM would remove those gains, which would be a big reason to not use this method. Can you comment on this?
>
> __Long Video vs Short Video__
>
> I still don't really see why this method is tailored for long video at all, and Reviewer 7bwU agrees. Nothing about learnable VTM specifically is more effective than other methods with more frames . Furthermore, as I mentioned, you can train a model on Kinetics-400 with 64 frames as well and we should be able to notice some sort of improvement. What exactly about this method is specialized for long videos and thus merits not evaluating on Kinetics (where there are more baseline results)?

---

> > ### Author Response · Authors · 2024-08-11
> > **Response by Authors**
> >
> > Thank you for your positive response. Please find our follow-up response below.
> > ***
> >
> > > **Saliency**
> >
> > We agree with the reviewer. We will revise the draft as suggested.
> >
> > > **Training time**
> >
> > In Table below, we list the training time on the LVU dataset. The training time of the learnable VTM is much faster than that of the baseline network without token merging. Also, compared to the naive VTM, which is the straightforward application of ToMe to the baseline network, the learnable VTM takes a slightly longer time. However, it is not a huge gap, which significantly diminishes the benefits of the proposed algorithm.
> >
> > | No Merging | Naive VTM | Learnable VTM |
> > |------------|-----------|---------------|
> > | 7.2h       | 3.7h      | 4.0h          |
> >
> > > **Long video vs short video**
> >
> > Long-form video understanding tasks not only require to model more frames but also need to capture the complex spatiotemporal short-term and long-term dependencies of semantics. To facilitate this, the learnable VTM finds out important tokens among numerous in the video through the learned token partition and increases the influence of these tokens in the attention processes after merging. By doing so, it obtains better understanding than the ‘ToMe’ on the long-form video understanding tasks, as shown in the experimental results.
> >
> > As stated in L51-52, in this work, we aim to explore various video token merging methods on the long-video understanding datasets (LVU, Breakfast, and COIN) and to find out the effective video token merging technique for long-video understanding. Thus, we don’t use all types of video benchmarks for the evaluation. However, we explore various token merging methods (including ToMe) on the long-form video understanding benchmarks and also compete with recent SOTA methods for long video understanding.
> >
> > As the reviewer pointed out, the proposed algorithm can be evaluated on the short-video understanding benchmarks (such as Kinetics-400), even though it is out of the original scope of this paper. Please note that Kinetics-400, as pointed out in the “What Makes a Video a Video: Analyzing Temporal Information in Video Understanding Models and Datasets, CVPR 2018”, is a relatively simple dataset with low information density; it contains one simple action for each video and thus complex information across many frames are not required to obtain good classification results on this dataset. Therefore, the proposed algorithm may not yield meaningful performance gap over other token merging baselines on the Kinetics-400 dataset, because this dataset may be less sensitive to information loss caused by suboptimal token merging than the long video understanding datasets which contain enormous amount of complex information. Due to the time limitation, we may not be able to finish the experiment within the rebuttal period. However, we will include performance of short-video understanding benchmarks as the supplementary of this paper.
> >
> > ***
> > Thank you again for your time and effort for reviewing our paper. We do appreciate it. If you have additional comments, please let us know.

---

### Author Rebuttal · Authors · 2024-08-07

We would like to thank all reviewers for their time and efforts for providing constructive reviews. Also, we extend our thanks to the program and area chairs. We have faithfully responded to all comments below.

---

### Decision · Program_Chairs · 2024-09-25

**Decision:**

Accept (poster)

**Comment:**

As the Area Chair overseeing the review of Submission4154, I recommend acceptance for NeurIPS 2024. The paper introduces an innovative learnable Video Token Merging (VTM) method that dynamically adjusts token merging based on saliency, showing significant improvements on long-video classification tasks on datasets such as LVU, Coin, and Breakfast.

Reviewer sUxH praises the method’s approach to computational efficiency and memory optimization, which are critical for processing long video inputs. The exploration of different token merging strategies provides a clear understanding of the proposed method’s advantages.

Reviewer rFDe’s concerns about evaluation and comparison baselines were addressed effectively in the authors' rebuttal. Their commitment to include additional experiments and adapt their approach in response to feedback highlights the paper's potential and thoroughness.

Reviewer 7bwU’s comments on novelty and design considerations for video processing were met with detailed clarifications, underscoring the method’s relevance and adaptability for complex video understanding tasks.

The paper’s theoretical and practical contributions offer valuable insights into optimizing transformer networks for video analysis, making a substantial impact on the field. The positive adjustments and responses to reviewer feedback justify its acceptance at the conference.